# The Effect of Psychological Anxiety Caused by COVID-19 on Job Self-Esteem and Job Satisfaction of Airline Flight Attendants

**DOI:** 10.3390/ijerph19074043

**Published:** 2022-03-29

**Authors:** Yelin Shin, Jinyoung Olivia Choi, Sunghyup Sean Hyun

**Affiliations:** School of Tourism, Hanyang University, 222, Wangsimni-ro, Seongdong-gu, Seoul 04763, Korea; yealin93@naver.com (Y.S.); cjinoung@hanyang.ac.kr (J.O.C.)

**Keywords:** cabin crew, COVID-19, psychological anxiety factors, salary reduction, perceived infection risk, career stagnation, welfare benefits, social perception, employment insecurity, job self-esteem, job satisfaction

## Abstract

This study aimed to investigate how psychological anxiety caused by COVID-19 has influenced airline cabin crew job self-esteem and job satisfaction. A questionnaire based on prior research was developed to identify factors of psychological anxiety among cabin crews as a result of COVID-19. The survey sample was limited to current cabin crews who experienced leave of absence due to COVID-19, and questionnaires were distributed to 201 crew members from 15 February to 15 April 2021. As a result of the analysis, the hypothesis that salary reduction, career stagnation, social perception, and employment insecurity have a significant effect on job self-esteem and job satisfaction was supported, while perceived infection risk and benefit reduction were rejected. This study found that psychological anxiety caused by COVID-19 affected cabin crew’s self-esteem and job satisfaction. These findings could aid in the development of strategies for effective airline human resource management to prevent psychological anxiety from creating stress and negatively affecting work. Furthermore, since the alert for the emergence of new viruses will not be eased in the future, this study will prevent psychological anxiety among cabin crews to cause job self-esteem and job dissatisfaction.

## 1. Introduction

Infectious diseases have caused numerous social and economic changes in human history. Many studies on the social and economic effects of infectious diseases on individual lives have been conducted, and it has been found that infectious diseases contribute to negative psychological anxiety [1]. In December 2019, a new virus, known as coronavirus, emerged, which caused the COVID pandemic. Despite the quarantine measures, COVID-19 spread not only in China but globally, and the World Health Organization (WHO) declared it a pandemic on 11 March 2020. The anxiety felt by cabin crew members who had no choice but to provide face-to-face services in an enclosed aircraft after COVID-19 was real.

Owing to measures such as banning foreigners from entering the country, self-quarantine for two weeks, and temporary closure of tourist attractions to prevent the spread of COVID-19, international demand for flights in 2020 decreased by 91 percent compared to the previous year, and domestic demand decreased by 58 percent [2]. Most domestic and international airlines offered their employee the option to go on leave while receiving leave allowances with at least 50% salary cuts. Some airlines are even considering organizational changes such as M&A and downsizing, which is increasing anxiety among airline employees. The airline industry was confronted with organizational changes, job insecurity, and mass layoffs. In fact, in October 2020, domestic low-cost airline A, which was pursuing M&A, laid off 605 aviation workers, including flight attendants. The employees are experiencing significant anxiety due to infection risks in contact-type work [1] and career stagnation due to the indefinite delay of promotion [3]. The trends of reduced welfare benefits across all airlines undermine job satisfaction and negative social gaze, and employment insecurity from manpower restructuring also impedes organizational commitment [4].

This study aimed to examine whether psychological anxiety factors triggered by COVID-19 affected job self-esteem and job satisfaction among airline cabin crews, as well as hypothesize influential relationships based on previous research and suggest solutions to these problems. This study is also expected to be used as a reference for airlines’ human resource management measures in the event of a future infection disaster.

## 2. Literature Review and Hypotheses

### 2.1. Anxiety Psychology

Anxiety is a universal emotional disorder experienced by humans when a dangerous situation occurs or when a threatening scene is perceived [5]. The reason that such anxiety is recognized as a problem is that it can cause problems in a variety of situations, including interactions with others, in addition to simply feeling anxious [6,7]. According to previous research by Kim (2000) [8], 24 percent of respondents reported anxiety in social gatherings, 54 percent in interviews, and 8 percent in daily lives. Since a large proportion of people experience anxiety in their daily lives, it is necessary to investigate how anxiety caused by COVID-19 affects cabin crews.

### 2.2. Salary Reduction

Salary is an agreed-upon fixed amount, which means that in terms of compensation, the worker receives compensation in exchange for providing labor. Compensation is an exchange relationship formed in exchange for an employee’s contribution to the organization [9]. According to Reilly (2003) [10], when employees are properly compensated for their contributions to the company, it not only encourages them to work harder but also helps increase productivity. Previous researchers have stated that an employee’s performance and the resulting high salary are closely related [11,12].

Salary reduction occurs when a company reduces an employee’s current monthly or annual salary. Salary reductions, whether voluntary or involuntary, have been implemented as a means for businesses to increase organizational profits while keeping employees employed in the face of deteriorating business conditions [13]. From May to November 2020, Korean Air operated with only a bare minimum of personnel, with the remainder of the cabin crew on paid leave. Many flight attendants experience financial difficulties and anxiety as a result of unexpected leave of absence and the resulting significant salary cuts of more than 50 percent. Previous research has shown that salary reduction leads to employee turnover and lower productivity; therefore, it is hypothesized that salary reduction caused by COVID-19 leads to turnover and lower productivity, as well as low job self-esteem.

**Hypothesis** **1** **(H1).***Salary reduction, a psychological anxiety factor caused by COVID-19, will have a negative impact on flight attendants’ job self-esteem*.

### 2.3. Perceived Infection Risk

Risk perception occurs in a variety of areas, including war, climate change, terrorism, and personal factors, in addition to infectious diseases [14]. Since it is formed by an individual’s cognitive processes, it is not consistently the same for everyone, but is subject to differences in risk intensity and attributes based on individual experience and cognition [15] and has a significant influence on future decision-making [16]. The emergence of a danger in the immediate environment as well as an increased individual perception of the impending risk are critical factors that prompt the mobilization of personal resources in health-related decision-making. Anyone can have a perception of infection risk, but this can be especially acute for workers who are required to interact with a large number of people. Especially, due to the closeness of airplane seats and the small area in which people travel, air travel poses a significant risk of infection.

Airlines are making various efforts to prevent the spread of the coronavirus through contact with passengers on board by the cabin crew. Nonetheless, there have been cases of transmission to confirmed passengers and COVID-19 infection among flight attendants, raising concerns about in-flight virus infection. According to Yang (2010) [1], a previous study of infection anxiety among nurses in tuberculosis high-risk departments, 4.39 out of five questionnaire responses, “I am worried that the risk of tuberculosis infection will increase while working,” indicated a high level of infection anxiety. Although no previous research has been conducted on the perception of infection risk or infection anxiety among cabin crews, it is highly likely that cabin crew members are also concerned about and aware of the risk of infection while on duty, at a level comparable to nurses in high-risk departments in the case of COVID-19. It was hypothesized that flight attendants’ perception of infection risk has an effect on lowering job self-esteem, based on a previous study that perceived infection anxiety induces resignation and intention to leave.

**Hypothesis** **2** **(H2).***Perceived infection risk, a psychological anxiety factor caused by COVID-19, will have a negative impact on flight attendants’ job self-esteem*.

### 2.4. Career Stagnation

Workers put a lot of effort into promotion, and these efforts are directly related to the atmosphere of the organization and the morale of the workers. Companies must prioritize the management of a pool of people who can be promoted [17], and the issue of promotion is considered one of the most important aspects of corporate organizational management [18,19]. Career stagnation is a condition in which the possibility of vertical advancement within the organization is limited, or movement to a position requiring more responsibility is obstructed. Career stagnation occurs not only when promotion is possible but also when there is no possibility of more difficult work [20].

Many airlines, faced with mismanagement as a result of COVID-19, are looking for a way out through drastic restructuring such as asset sales and mergers and acquisitions. As part of their survival strategy, they indefinitely postponed their 2020 promotion announcements. Employees who had been preparing for promotion tests for several years, improving their foreign language scores and acquiring in-flight broadcasting qualifications for promotion, were extremely disappointed over the delay in the promotion announcement. When employees experience career stagnation, individual job dissatisfaction increases, and as a result, job performance decreases [21]. Prior studies suggested that the higher the perception of career stagnation, the lower the job satisfaction level, leading to the hypothesis that career stagnation caused by COVID-19 lowers cabin crews’ job self-esteem and job satisfaction.

**Hypothesis** **3** **(H3).**
*Career stagnation, a psychological anxiety factor caused by COVID-19, will have a negative impact on flight attendants’ job self-esteem.*


### 2.5. Reduced Welfare Benefits

Lee [22] defined welfare as indirect compensation other than workers’ basic wages, allowances, and bonuses, as well as a social structure system designed to increase workers’ productivity and improve job stability. Rather than being determined by law, the concept of welfare is generally based on the results of an agreement between labor and management [23]. Workers are interested in how well the welfare benefits that companies invest in are balanced [24], and workers who believe these benefits are appropriate feel satisfied with the organization and immersed in their jobs. Previous studies on welfare benefits have been conducted in a variety of occupations, and according to Sung ’s (2019) study on the welfare benefits of hotel companies [25], welfare benefits had a significant effect on employee job satisfaction.

Airlines, like other businesses, provide various benefits to their employees. The most significant benefit of airlines is their ability to purchase tickets at low cost. However, due to the risk of infection from COVID-19, most countries worldwide have implemented mandatory quarantine measures for more than two weeks, making it virtually impossible for flight attendants to enter and exit for the purpose of travel. As a result, employee flight ticket benefits, one of the most important welfare benefits for flight attendants, have become obsolete. Previous research found that satisfactory welfare benefits had a positive effect on job satisfaction and organizational commitment. Thus, it was hypothesized that reducing cabin crew members’ welfare benefits due to COVID-19 would have a negative impact on job self-esteem and job satisfaction.

**Hypothesis** **4** **(H4).***Reduced welfare benefits, a psychological anxiety factor caused by COVID-19, will have a negative impact on flight attendants’ job self-esteem*.

### 2.6. Social Perception

Social perception refers to the interpretation of social situations or states by discriminating between organized groups using people’s sensory intuition. That is, the perception of a specific job and group is influenced by the authority, respect, value, and importance that members of society have in general [26]. Social perception is also closely related to symbolic interaction theory, in that individuals evaluate themselves based on observations and interpretations gleaned from interactions with others, and they value how others perceive or evaluate them [27]. Individuals who have experienced a decrease in their value as a result of interaction with others, that is, individuals who believe they are perceived negatively by others, suffer from low self-esteem and negative self-judgement [28].

Recently, due to COVID-19, the social perception of cabin crew members who frequently travel abroad has been negative. Due to the occupational characteristics of mandatory international travel, it is viewed not only as a medium for the inflow of overseas infections but also as a potential confirmed case, so that the cabin crew’s family and acquaintances must refrain from using multiple facilities (kindergartens, schools, academies, etc.) Some flight attendants who are burdened by such negative social perceptions and prejudices apply for long-term unpaid leaves of absence or even consider resigning. It was hypothesized that the higher the social perception of airlines or flight attendants, the more positively would the job self-esteem of flight attendants be impacted.

**Hypothesis** **5** **(H5).***Social perception,**a psychological anxiety factor caused by COVID-19, will have a positive impact on flight attendants’ job self-esteem*.

### 2.7. Employment Insecurity

Halier and Lyon (1996) defined employment insecurity as a worker’s assessment of the likelihood of future job loss [29]. Workers experience a sense of helplessness as a result of these threats, which reduces their job enthusiasm [30]. The greater the workers’ involvement in their jobs and their trust in organizational commitments, the more threatened they feel [31]. Thus, threats such as perceived employment insecurity or unemployment are likely to cause mental distress [32].

International flights were reduced by approximately 91% as a result of COVID-19, while domestic flights were reduced by 58%. Korean Air, as an example of a domestic airline, began a circular leave of absence. The government has provided subsidies on the condition of employment security to provide leave allowances to employees on leave of absence (as of October 2020). However, it is difficult to predict the tenure of subsidy. If the government refuses it in the future, employees’ employment stability will be jeopardized. Most airline employees experience employment insecurity as a result of COVID-19, some of whom also consider leaving. It has been hypothesized that concerns about cabin crew employment insecurity caused by COVID-19 will negatively affect job self-esteem.

**Hypothesis** **6** **(H6).***Employment insecurity,**a psychological anxiety factor caused by COVID-19, will have a negative impact on flight attendants’ job self-esteem*.

### 2.8. Job Self-Esteem

Self-esteem is a general assessment of one’s own abilities and the degree to which one considers oneself competent and necessary [33]. Job self-esteem refers to self-esteem that comes from the organization, in which workers consider their own valuation [34]. This is due to organizational activities within the job, such as the process of workplace communication, the act of the organization evaluating the value of workers, and quality of work life.

When employees have a positive attitude toward and a strong sense of organizational identity, their job self-esteem rises. According to a study conducted by Kim [35], employees experience job self-esteem when they receive positive evaluations within the organization or when they are recognized as essential to the organization. Job self-esteem increases when one’s role or job is deemed important. Furthermore, the more satisfied the working environment, the higher the employee’s self-esteem. Consequently, job self-esteem can be viewed as an expanded form of job satisfaction, which is a qualitative component of work life and should be regarded as an important factor in working life.

### 2.9. Job Satisfaction

Job satisfaction refers to employees’ positive psychological state regarding their jobs [36] and can manifest differently depending on their work position, personal growth, age, and salary [37]. Robert and Meyer defined job satisfaction as an emotional commitment to the job as a whole or to a specific aspect [38]. Emotional commitment is defined as the degree to which an individual identifies with and participates in an organization. Employees are satisfied with their jobs when they believe they are properly compensated, treated, promoted, and cared for within the organization [39]. Job satisfaction also has a significant social value. The workplace is a place where they can spend the majority of their days pursuing rewards and satisfaction, which are closely related to their quality of life. Furthermore, dissatisfaction with one’s job has a negative impact on home and leisure life due to the transfer effect, as well as physical health due to stress [40].

Job satisfaction is related to the performance of organizational members, and employees with high job satisfaction have lower turnover and absenteeism rates as well as a high level of organizational citizenship behavior [41]. Furthermore, people who are satisfied with their jobs strive to maximize their ability to achieve the organization’s goals. This means that job satisfaction improves organizational commitment and has a significant impact on an organization’s success. Job satisfaction should be regarded as a critical factor because it not only benefits individuals but also improves organizational performance. An airline cabin crew member has a job that provides close service to customers, and if the job satisfaction increases, so will the quality of service, leading to customer satisfaction. Previous research on the relationship between job self-esteem and job satisfaction suggests that flight attendants with low job self-esteem have low job satisfaction.

**Hypothesis** **7** **(H7).***Job self-esteem will have a positive impact on flight attendants’ job satisfaction*.

## 3. Research Methods

### 3.1. Research Model and Hypotheses

This study aimed to investigate the impact of six psychological anxiety factors caused by COVID-19 on airline flight attendants’ self-esteem and job satisfaction. The research model was developed on the basis of the theoretical background of the literature review, as shown in Figure 1.

### 3.2. Measurement

The questionnaire used in this study was developed by selecting measurement items based on operational definitions of the variables described above. Among the psychological anxiety factors caused by COVID-19, salary reduction was measured using three items including falling salary levels and dissatisfaction with salary structure and management [42]; perceived infection risk was measured using four items based on concerns about the risk of infection during flight work [43]; career stagnation was assessed using three items including structural and content career plateaus. The reduced welfare benefits were measured using three items [25]; four items were used to assess social perception, including the perception of negative social views on flight attendants following COVID-19 and the situations they have faced as a result [44]; and employment insecurity was measured using three items [29]. Furthermore, job self-esteem consists of four items, including pride in one’s job and company and a sense of accomplishment [45]. The dependent variable, job satisfaction, consisted of four items. All measurement items were assessed using a 5-point Likert scale ranging from 1 (not at all) to 5 (very much).

### 3.3. Data Collection and Analysis Method

The survey was conducted over two months from 15 February to 15 April 2021 by distributing 201 questionnaires to currently-employed cabin crew members who had taken leave due to COVID-19. The questionnaire was distributed both offline and online, and 201 effective responses were analyzed.

The statistical analysis methods used to analyze the collected data were as follows: First, we conducted frequency analysis to understand the demographic characteristics of the participants. Second, we used a confirmatory factor analysis (CFA) to assess the validity of the latent and observed variables. Convergent and discriminant validity were verified using composite reliability (CR) and average variance extracted (AVE). In addition, the reliability of the measurement was confirmed using Cronbach’s α. Third, descriptive statistics were analyzed to identify the levels of psychological anxiety factors, job self-esteem, and job satisfaction of participants and to determine whether normality assumptions of the collected data were met. Fourth, a structural equation model (SEM) was used to verify the causal relationships between psychological anxiety factors, job self-esteem, and job satisfaction. Finally, we used bootstrapping to test the indirect effects of psychological anxiety factors on job satisfaction via job self-esteem. SPSS 25 and AMOS 25 programs were used for statistical analysis.

## 4. Research Results

### 4.1. Demographic Characteristics of Participants

Table 1 presents the demographic profiles of the survey participants. In terms of sex, 171 (85.1%) were women and 30 (14.9%) were men; the proportion of women was much higher. In terms of participants’ ages, those aged 26–30 (41.8%) were the most numerous, followed by those aged 31–35 (36.3%). Unmarried participants (65.7%) were approximately twice as many as married (34.3%), and the majority of respondents had a university degree (79.1%). The participants were 124 flight attendants (61.7%), 52 assistant pursers (25.9%), 15 pursers (7.5%), 9 internship flight attendants (4.5%), and 1 senior purser (0.5%). More than half of the participants had worked for more than five years but less than ten years, with a working year average of 6.83. As for annual income, 40–50 million won was the highest with 92 people (45.8%), followed by 43 people earning 50–60 million won (21.4%), 37 people earning 30–40 million won (18.4%), 23 people earning 60–70 million won (11.4%), and 6 people earning more than 70 million won (3.0%).

### 4.2. Confirmative Factor Analysis and Reliability Analysis

Prior to proceeding with the structural equation model analysis, confirmatory factor analysis was conducted to ensure that the observed variables constituting each latent variable were correctly constructed. The results of the analysis are presented in Table 2. The factor loadings of all items ranged between 0.530 and 0.914, which was greater than the criterion of 0.50. Reliability analysis was also performed to confirm the internal consistency of the measurement items using Cronbach’s α reliability index. The Cronbach’s α coefficients of all factors were confirmed to be 0.70 or higher, indicating that the measurement tool used in this study had good internal consistency.

The fit of the measurement model is considered acceptable when the Comparative Fit Index (CFI) and Tucker–Lewis Index (TLI) are 0.90 or higher, and Root-Mean Square Error of Approximation (RMSEA) is less than 0.08 (Bae, 2014). In this measurement model, CFI was 0.939, TLI was 0.928, both of which were above the criteria, and RMSEA was 0.058, which also met the criterion for goodness of fit.

### 4.3. Convergent Validity and Discriminant Validity Verification

Convergent validity, which evaluates whether the observed variables are sufficiently correlated to converge well with each other, and discriminant validity, which assesses whether the latent variables are independent concepts without excessive similarity between variables, are both required. Convergent validity was verified using CR and AVE, and the results are shown in Table 3. In general, convergent validity is considered good when the CR is 0.70 or higher, and the AVE is 0.50 or higher. The results of the analysis showed that the CR of all variables was greater than 0.70, and the AVE was greater than 0.50, indicating that the convergent validity of the measurement model was excellent.

The results of the correlation analysis are presented in Table 4. Table 5 presents the results of discriminant analysis. Job self-esteem had a statistically significant negative correlation with salary reduction, perceived infection risk, career stagnation, reduced welfare benefits, and employment insecurity and a positive correlation with social recognition. Job self-esteem and satisfaction showed a significant positive correlation.

The discriminant validity test proposed by Bagozzi and Yi (1988) was used to confirm whether the high correlation between variables inhibited the discriminant validity of the measurement model. This method compares the chi-square statistics of the “Merge model,” which combines job self-esteem and job satisfaction with the highest correlation into a single factor, and the “Original model,” which separates job self-esteem and job satisfaction.

The original model had a chi-square statistic of 535.334 and a degree of freedom of 322, whereas the merged model had a chi-square statistic of 568.755 and a degree of freedom of 329. When the results of the two models were compared, the chi-square statistics differed by 33.421 and the degrees of freedom differed by 7. When the number of degrees of freedom was 7, the chi-square statistic difference of 33.421 between the two models was greater than the threshold of 14.067. The original model for separating job self-esteem and job satisfaction was confirmed better. The goodness-of-fit indices such as CFI, TLI, and RMSEA were also analyzed, and the indices of the original model were found to be superior. Consequently, the discriminant validity of the measurement model in this study was ensured.

### 4.4. Descriptive Statistics and Normality Verification

Descriptive statistics were analyzed to determine the levels of psychological anxiety factors caused by COVID-19, job self-esteem, and job satisfaction, and the results are presented in Table 6. The results showed that the average salary reduction was 4.08, perceived infection risk was 3.50, career stagnation was 3.56, reduced welfare benefits were 3.70, social perception was 3.30, and employment insecurity was 3.95. Flight attendants had the highest level of anxiety about salary reduction, and employment insecurity was also relatively high. The average job self-esteem as a mediating variable was 3.37, and average job satisfaction as a dependent variable was 3.02.

Skewness and kurtosis were identified to check whether the collected data met the normality assumption. If skewness is less than ±2 and kurtosis is less than ±7, the data can be judged to satisfy the normal distribution assumption (Curran, West, and Finch, 1996). Because all variables are found to be less than the reference values, the data distribution configured for structural equation model analysis is satisfactory.

### 4.5. Structural Model Analysis and Hypothesis Verification

#### 4.5.1. Model Fit Analysis

Model fix indices were identified to assess the fit of the research model, and the results are presented in Table 7. The CFI, TLI, and RMSEA indices were used in the analysis. It was found that CFI was 0.937, TLI was 0.928, and RMSEA was 0.058, all of which met the criteria. As a result, the fit of the structural equation model was judged adequate.

#### 4.5.2. Direct Effects Analysis Results

Table 8 presents the results of the SEM’s direct effect between the variables. The values of the path coefficients between the variables were as follows: Salary reduction, career stagnation, and employment insecurity had significant negative effects on job self-esteem, with path coefficient values of −0.188 (*p* < 0.05), −0.249 (*p* < 0.01), and −0.264 (*p* < 0.01), respectively. Social perception had a significant positive effect on job self-esteem, with a path coefficient value of 0.189 (*p* < 0.05). In other words, the greater the anxiety about salary reduction, career stagnation, and employment insecurity, the lower the self-esteem of the job. The better the social perception, the higher the self-esteem of the job.

Furthermore, the direct path from job self-esteem to job satisfaction yielded a significantly positive result (Î^2^ =0.941, *p* < 0.001). It can be concluded that the higher one’s self-esteem, the greater one’s job satisfaction.

#### 4.5.3. Indirect Effects Analysis

The bootstrapping method was used to validate the indirect effects of job self-esteem on the relationship between six psychological anxiety factors caused by COVID-19 and job satisfaction; the results are shown in Table 9. The sample size for bootstrapping was set at 2000, and statistical significance was verified at the 95% confidence level. The results showed that salary reduction had a negative effect on job satisfaction through job self-esteem (β = −0.177, *p* < 0.05), and career stagnation also had a negative effect on job satisfaction through job self-esteem (β = −0.234, *p* < 0.05). Social perception had a positive effect on job satisfaction through job self-esteem (β = 0.178, *p* < 0.05). Employment insecurity was also found to have a negative effect on job satisfaction through job self-esteem (β = −0.249, *p* < 0.05).

#### 4.5.4. Summary of SEM Analysis

Table 10 summarizes the structural equation model analysis results of this study. Figure 2 presents the structural equation model analysis results. It was verified that salary reduction, career stagnation, and employment insecurity reduced cabin crew members’ job self-esteem, whereas social perceptions increased job self-esteem. Furthermore, job self-esteem has been shown to increase job satisfaction. Thus, Hypotheses 1, 3, 5, 6, and 7 were supported.

## 5. Conclusions

Due to the prolonged COVID-19 pandemic, airline cabin crews are experiencing a variety of challenges, including unpaid leave and salary reductions due to intermittent flight work. This study aimed to investigate the impact of psychological anxiety factors felt by cabin crews on airline difficulties caused by COVID-19 on job self-esteem and job satisfaction. A structural equation model analysis was performed to confirm the causal relationships between six psychological anxiety factors (salary reduction, perceived infection risk, career stagnation, reduced welfare benefits, social perception, and employment insecurity), job self-esteem, and job satisfaction. Previous studies have primarily focused on the effects of organizational commitment on the job self-esteem and job satisfaction of cabin crews and organizational culture. The majority of them, in particular, were based on intrinsic factors originating within the airline organizations, and studies on psychological anxiety factors caused by external factors, such as COVID-19, were insufficient. It is critical to manage the psychological anxiety of organizational members because it lowers morale at work and negatively impacts business. The findings of this study could aid in the development of ways to improve the job self-esteem and job satisfaction of cabin crews suffering psychological anxiety caused by COVID-19. The results and implications of this study are as follows.

First, salary reduction was found to be detrimental to cabin crew self-esteem (β =−0.188, C.R. = −2.204, *p* < 0.028). In other words, cabin crew members experience anxiety when their salaries are reduced due to COVID-19, and this anxiety affects their self-esteem. Maintaining flights with only a bare minimum of manpower, forcing cabin crew members to take unpaid leave, and a more than 50% reduction in actual salaries cause economic difficulties and anxiety for many flight attendants. The wage mechanism plays a key role between employers and employees [46] and helps increase productivity when employees believe they are being fairly compensated [10]. As demonstrated by previous scholars, it was confirmed that lowering cabin crew salaries has a significant negative impact on job self-esteem. Airlines should create a system that can maintain the salary system to some extent, even if employees take leave of absence in case of various disasters or pandemics such as COVID-19, to prevent employees from falling into job dissatisfaction.

Second, Hypothesis 2, which proposed that the airline cabin crew’s perception of infection risk would have a negative impact on their self-esteem, was rejected. The results of this study survey showed a high level of concern among cabin crews about being infected while flying. The response rate was high, with 3.8 out of 5 respondents saying they were worried that passengers infected with COVID-19 might be flight passengers they serve, and 3.52 out of 5 respondents questioned whether there was a risk of infection due to proper wearing of protective equipment. This was the result of a similar response to a study on the perception of infection anxiety among nurses in a high-risk department for tuberculosis [1]. Cabin crew members had a high perception of infection anxiety while on duty; however, such a perception had no significant impact on their job self-esteem (β = −0.018, C.R. = −0.200, *p* < 0.841). To lower the risk of infection among flight attendants, it is necessary to guide them to wear appropriate protective equipment and change services to minimize direct contact with passengers.

Third, career stagnation was found to have a negative impact on cabin crew job self-esteem (β = −0.249, C.R. = −2.860, *p* < 0.004). This is consistent with research findings that career stagnation reduces organizational attachment (Park, 2020). This is also consistent with a study conducted by Seo (2012), who found that the greater the perception of career stagnation, the lower the job satisfaction. In other words, when cabin crew members’ careers stagnate due to COVID-19, they experience anxiety, which affects their job self-esteem. Since the cabin crew’s career stagnation negatively affects organizational commitment, airlines are trying to lower the crew’s turnover intention [47]. There is a need for appropriately timed promotion announcements and plans to reduce career stagnation. Providing a new environment on a regular basis to provide anticipation to cabin crew members whose promotions have stalled would be an effective way for an organization to alleviate career stagnation.

Fourth, Hypothesis 4, which predicted that the reduced welfare benefits caused by COVID-19 would have a negative impact on job self-esteem, was rejected. According to previous studies, corporate welfare benefits improve employees’ job satisfaction and quality of life [47]. However, this study revealed that the reduction in welfare benefits had no significant impact on cabin crews’ job self-esteem (β = 0.024, C.R. = 0.318, *p* < 0.750). Benefits, as additional compensation, do not appear to be a major consideration for flight attendants in situations in which the most important decision is whether to quit or wait for things to improve.

Fifth, the effect of social perception on job self-esteem of cabin crews was positive (β = 0.189, C.R. = −2.054, *p* < 0.040). Individuals are sensitive to changes in the social environment of the organization to which they belong [28], and the social perception of the organization has a significant impact on the formation of an individual’s self. When flight attendants’ social perception worsens, as is the case with the recession and restructuring of the airline industry as a result of COVID-19, and when they are considered as potential infection carriers due to the fact of working abroad frequently, they feel anxiety, which has an impact on job self-esteem. Airlines will have to come up with a way to improve the image of the airline and its employees for flight attendants to restore their social perception lost due to COVID-19. Marketing efforts are especially needed to shift the existing perception of flight attendants, which were only evaluated as a simple service job, to include a sense of duty to take responsibility for passengers’ safety and take on the weight of the job.

Sixth, it was found that employment insecurity, a psychological anxiety factor caused by COVID-19, had a negative effect on cabin crews’ job self-esteem (β = −0.264, C.R. = −2.760, *p* < 0.006). Due to the economic downturn in the airline industry as a result of the COVID-19 pandemic, airline employees are being restructured, and employees are taking leave of absence. The perception of employment insecurity among cabin crews was high enough, with an average of 4.13 out of 5, when asked if they believed there was a high likelihood of layoffs in the future. Organizations should not overlook the effects of employment insecurity and uncertainty caused by organizational change to ensure job stability for aviation industry workers who are suffering from severe job insecurity due to COVID-19. It is critical to reduce anxiety factors ahead of time by providing information and clear notice of changes due to organizational change and manpower reduction plans. Furthermore, the government should prepare long-term support measures rather than temporary and short-term assistance and should concentrate on implementing airline relief measures with strategic and effective financial assistance.

Finally, an analysis of the indirect effects of psychological anxiety factors caused by COVID-19 on job satisfaction through mediating job self-esteem revealed that salary reduction, career stagnation, and employment insecurity had a negative effect on job satisfaction through job self-esteem, whereas social perception was found to have a positive effect. Airlines will need to develop human management programs, such as employee training and psychological counseling, to alleviate flight attendants’ depression and anxiety when the working environment and conditions change due to unexpected environmental changes such as COVID-19.

## 6. Limitation and Future Research

The limitations of this study and suggestions for future research are as follows. Since the survey in this study is limited to large domestic airlines, the expansion and application of the findings to cabin crews of foreign airlines or other domestic airlines has not been possible. To generalize the findings of this study, it is necessary to expand the sample to include low-cost airlines and foreign airline cabin crews. Furthermore, this study was conducted at a time when the passenger transportation industry was facing a serious crisis as a result of the COVID-19 shutdown and restrictions on cross-border movement. A more accurate and rigorous follow-up study should be conducted by dividing the psychological anxiety factors experienced by airline cabin crews before and after the COVID-19 pandemic, using a methodology that controls the timing of anxiety.

## Figures and Tables

**Figure 1 ijerph-19-04043-f001:**
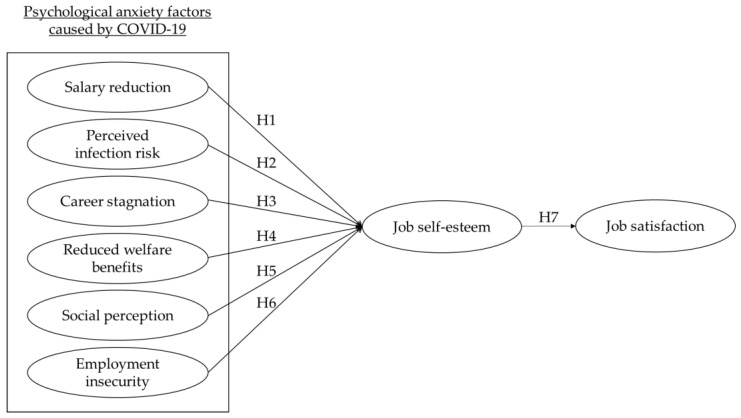
Research model.

**Figure 2 ijerph-19-04043-f002:**
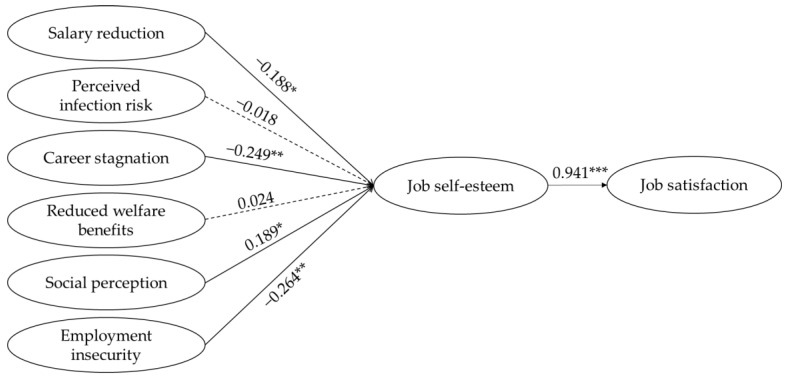
Structural equation model analysis results. Note: * *p* < 0.05, ** *p* < 0.01, *** *p* < 0.001.

**Table 1 ijerph-19-04043-t001:** Demographic characteristics of participants.

Variable	Index	Frequency (*n*)	Percent (%)	Mean (SD)
Gender	Female	171	85.1	
	Male	30	14.9	
Age	20–25	12	6.0	
	26–30	84	41.8	
	31–35	73	36.3	
	36–40	26	12.9	
	41–45	5	2.5	
	over 46	1	0.5	
Marital Status	Unmarried	132	65.7	
	Married	69	34.3	
Education	College degree	30	14.9	
	University degree	159	79.1	
	In graduate school	11	5.5	
	Graduate degree	1	0.5	
Working Period	Less than 5 years	46	22.9	6.83 (3.83)
	5–10 years	120	59.7	
	Over 15 years	35	17.4	
Working Position	Flight attendant(internship)	9	4.5	
	Flight attendant	124	61.7	
	Assistant Purser	52	25.9	
	Purser	15	7.5	
	Senior Purser	1	0.5	
Average Annual Income	KRW 30–40 million	37	18.4	
	KRW 40–50 million	92	45.8	
	KRW 50–60 million	43	21.4	
	KRW 60–70 million	23	11.4	
	Over KRW 70 million	6	3.0	
Total		201	100.0	

**Table 2 ijerph-19-04043-t002:** Confirmative factor analysis and reliability analysis results.

Factors	Items	M (SD)	Loading	Cronbach’s α
Salary reduction (3)	I have developed economic anxiety as a result of salary reduction caused by COVID-19.	4.14 (0.84)	0.730	0.775
	I am struggling to reduce my basic living expenses due to a salary reduction following COVID-19.	3.88 (0.95)	0.725	
	I am concerned that the reduced salary will be extended.	4.22 (0.88)	0.750	
Perceived infection risk (4)	I am concerned that COVID-19-infected passengers may be on the same flight.	3.80 (0.76)	0.776	0.854
	I have considered leaving my job as a flight attendant and looking for another because of the risk of infection.	3.26 (0.95)	0.724	
	It is debatable whether wearing protective equipment reduces the risk of infection.	3.52 (0.83)	0.822	
	For a variety of reasons, I am concerned that I will be more susceptible to COVID-19 infection.	3.41 (0.86)	0.785	
Career stagnation (3)	I am concerned that I will not be promoted, despite the fact that the possible year has passed.	3.77 (0.71)	0.717	0.848
	I do not find my current position to be challenging or interesting.	3.45 (0.84)	0.837	
	I think I am losing focus on what I should do to advance my career.	3.47 (0.82)	0.864	
Reduced welfare benefits (3)	I am concerned that company’s welfare benefits will be reduced due to COVID-19.	3.64 (0.79)	0.828	0.853
	I do not seem to be using welfare benefits as frequently after COVID-19.	3.89 (0.85)	0.758	
	If the company’s welfare benefits are reduced, I am less likely to stay motivated on my job.	3.58 (0.82)	0.858	
Social perception (4)	I believe the social perception of flight attendants has deteriorated since COVID-19.	2.68 (0.97)	0.857	0.874
	My job appears to have been less desirable than before, after COVID-19.	2.68 (0.94)	0.814	
	I am hesitant to tell others about my job after COVID-19.	3.01 (1.03)	0.779	
	I have been unable to use hospitals or other external facilities due to my job as a flight attendant since COVID-19.	2.42 (0.92)	0.743	
Employment insecurity (3)	I believe I may be forced to leave the company against my will, following the COVID-19 situation.	4.03 (0.99)	0.834	0.836
	I believe there is a high likelihood of redundancy in the future, following the COVID-19 situation.	4.13 (0.88)	0.802	
	I have been considering resignation or a change of jobs due to employment insecurity since COVID-19.	3.67 (1.21)	0.791	
Job self-esteem (4)	I can proudly tell those around me that I am a flight attendant.	3.32 (1.04)	0.777	0.852
	I am honored to be a part of this company.	3.16 (1.17)	0.834	
	I believe that taking pride in my work contributes to the development of my personal self-esteem and a sense of accomplishment.	3.77 (0.91)	0.530	
	My job is enjoyable and interesting to me.	3.24 (1.07)	0.914	
Job satisfaction (4)	I feel proud of what I do.	3.27 (0.99)	0.908	0.915
	I would like to strongly recommend my job.	2.92 (1.06)	0.862	
	I am currently interested in and enjoy my job.	3.15 (1.03)	0.902	
	I am satisfied with the working environment at my company.	2.76 (0.99)	0.745	

χ^2^ = 535.334 (df = 322, *p* < 0.001), CFI = 0.939, TLI = 0.928, RMSEA = 0.058.

**Table 3 ijerph-19-04043-t003:** Convergent validity analysis result.

Variables	Composite Reliability (CR)	Average Variance Extracted (AVE)
Salary reduction	0.817	0.598
Perceived infection risk	0.893	0.675
Career stagnation	0.904	0.759
Reduced welfare benefits	0.971	0.919
Social perception	0.884	0.657
Employment insecurity	0.840	0.637
Job self-esteem	0.852	0.598
Job satisfaction	0.914	0.729

**Table 4 ijerph-19-04043-t004:** Correlation analysis result.

Variables	1	2	3	4	5	6	7	8
1. Salary reduction	1							
2. Perceived infection risk	0.279 **	1						
3. Career stagnation	0.349 ***	0.569 ***	1					
4. Reduced welfare benefits	0.149	0.333 ***	0.271 **	1				
5. Social perception	−0.278 **	−0.508 ***	−0.424 ***	−0.491 ***	1			
6. Employment insecurity	0.515 ***	0.459 ***	0.361 ***	0.303 ***	−0.540 ***	1		
7. Job self-esteem	−0.455 ***	−0.411 ***	−0.453 ***	−0.249 **	0.491 ***	−0.558 ***	1	
8. Job satisfaction	−0.452 ***	−0.414 ***	−0.539 ***	−0.238 **	0.451 ***	−0.513 ***	0.935 ***	1

** *p* < 0.01, *** *p* < 0.001.

**Table 5 ijerph-19-04043-t005:** Discriminant analysis result.

Model	χ^2^	df	Δχ^2^	Δdf	CFI	TLI	RMSEA
Original Model	535.334	322	33.421	7	0.939	0.928	0.058
Merge Model	568.755	329			0.931	0.921	0.060

**Table 6 ijerph-19-04043-t006:** Descriptive statistics analysis result.

Variables	Range	Average	Standard Deviation	Skewness	Kurtosis
Salary reduction	1–5	4.08	0.74	−1.32	2.12
Perceived infection risk	1–5	3.50	0.71	−0.63	1.01
Career stagnation	1–5	3.56	0.69	−0.68	0.89
Reduced welfare benefits	1–5	3.70	0.72	−0.79	1.26
Social perception	1–5	3.30	0.82	−0.49	0.29
Employment insecurity	1–5	3.95	0.90	−0.85	0.37
Job self-esteem	1–5	3.37	0.88	0.02	−0.58
Job satisfaction	1–5	3.02	0.91	0.18	−0.35

**Table 7 ijerph-19-04043-t007:** Structural equation model’s fit analysis result.

χ^2^	df	*p*	CFI	TLI	RMSEA
546.140	328	<0.001	0.937	0.928	0.058

**Table 8 ijerph-19-04043-t008:** Direct effects analysis result.

Direct Path	B	SE	β	C.R.	*p*
Salary reduction → Job self-esteem	−0.247	0.112	−0.188	−2.204 *	0.028
Perceived infection risk → Job self-esteem	−0.021	0.104	−0.018	−0.200	0.841
Career stagnation → Job self-esteem	−0.391	0.137	−0.249	−2.860 **	0.004
Reduced welfare benefits → Job self-esteem	0.029	0.091	0.024	0.318	0.750
Social perception → Job self-esteem	0.183	0.089	0.189	2.054 *	0.040
Employment insecurity → Job self-esteem	−0.257	0.093	−0.264	−2.760 **	0.006
Job self-esteem → Job satisfaction	1.073	0.086	0.941	12.458 ***	0.000

* *p* < 0.05, ** *p* < 0.01, *** *p* < 0.001.

**Table 9 ijerph-19-04043-t009:** Indirect effects analysis result.

Indirect Path	β	SE	95% CI	*p*
Salary reduction → Job satisfaction	−0.177 *	0.104	−0.348~0.000	0.049
Perceived infection risk → Job satisfaction	−0.017	0.087	−0.160~0.121	0.419
Career stagnation → Job satisfaction	−0.234 *	0.109	−0.424~−0.061	0.010
Reduced welfare benefits → Job satisfaction	0.022	0.082	−0.110~0.156	0.385
Social perception → Job satisfaction	0.178 *	0.095	0.023~0.334	0.031
Employment insecurity → Job satisfaction	−0.249 *	0.123	−0.461~−0.049	0.021

* *p* < 0.05.

**Table 10 ijerph-19-04043-t010:** Summary of research model verification result.

Hypotheses	Result	Standardized Coefficient (β)
H1. Salary reduction → Job self-esteem	Support	−0.188
H2. Perceived infection risk → Job self-esteem	Not Support	−0.018
H3. Career stagnation → Job self-esteem	Support	−0.249
H4. Reduced welfare benefits → Job self-esteem	Not Support	0.024
H5. Social perception → Job self-esteem	Support	0.189
H6. Employment insecurity → Job self-esteem	Support	−0.264
H7. Job self-esteem → Job satisfaction	Support	0.941

## Data Availability

Data sharing is not applicable to this article.

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
