# Peer review of "The Effect of Psychological Anxiety Caused by COVID-19 on Job Self-Esteem and Job Satisfaction of Airline Flight Attendants"

_ijerph, 2022, doi:10.3390/ijerph19074043_

Round 1
Reviewer 1 Report
To begin, you did not properly motivate this study from a practical standpoint.
Why is this topic important right now, especially in light of the current competitive conditions?
What kinds of industry-specific evidence can you offer that would make industry leaders sit up and pay attention to your study?
Answers to these and related questions will help make a much stronger case for pursuing this line of inquiry.
Additionally, you did not properly motivate this study from a conceptual standpoint.
What are the specific gaps in the literature that you are attempting to address? I encourage you to clearly and cogently explain:
(1) how the focal study addresses an important industry priority;
(2) how the topic is connected to existing theory;
(3) what we already know about practice and theory;
(4) what specifically we do not know;
(5) why we need to know what we do not know;
and (6) how this study or inquiry will help close the practical and theoretical gaps between what we know and do not know.
Reviewer 2 Report
Thanks to the Authors for the opportunity to see their work, I leave my comments for consideration below. First of all, congratulations to the Authors of a well-written article.
- Literature Review and Hypotheses
The theoretical part is concise and very well written. I suggest authors to consider including the most important information from the following articles:
https://doi.org/10.3390/su13179704
https://doi.org/10.3390/ijerph18073419
https://doi.org/10.3390/ijerph18073383
- Research Methods
3.2. Measurement
At this point, I am asking the Authors to describe in more detail the research tools used
- Research Results
Everything that is needed is presented in the main part describing the results of the research. The results were presented in what I believe is the best. At the beginning, demographic data were presented, and then the next steps of statistical analysis. Because in my articles I use similar methods myself. I don't think that this part would need to be changed.
- Discussion and Implications
I do not think that in the present form of the article I can propose anything that will improve its quality. The article is well written
- Limitation and Future Research
In the limitations of the research, it is worth mentioning that the research refers to a situation in which the limitations of flying were greater than the present ones
